# Ion Transporters, Channelopathies, and Glucose Disorders

**DOI:** 10.3390/ijms20102590

**Published:** 2019-05-27

**Authors:** Huseyin Demirbilek, Sonya Galcheva, Dogus Vuralli, Sara Al-Khawaga, Khalid Hussain

**Affiliations:** 1Department of Paediatric Endocrinology, Hacettepe University Faculty of Medicine, 06230 Ankara, Turkey; dr_huseyin@hotmail.com (H.D.); dvuralli@hotmail.com (D.V.); 2Department of Paediatrics, Varna Medical University/University Hospital “St. Marina”, Varna 9002, Bulgaria; sonya_galcheva@abv.bg; 3Department of Paediatric Medicine, Division of Endocrinology, Sidra Medicine, Doha, Qatar; SarAlKhawaga@hbku.edu.qa; 4College of Health & Life Sciences, Hamad Bin Khalifa University, Qatar Foundation, Education City, Doha, Qatar

**Keywords:** beta cell, K_ATP_ channel, voltage-gated calcium channels, membrane transporters, neonatal diabetes, hyperinsulinaemic hypoglycaemia

## Abstract

Ion channels and transporters play essential roles in excitable cells including cardiac, skeletal and smooth muscle cells, neurons, and endocrine cells. In pancreatic beta-cells, for example, potassium K_ATP_ channels link the metabolic signals generated inside the cell to changes in the beta-cell membrane potential, and ultimately regulate insulin secretion. Mutations in the genes encoding some ion transporter and channel proteins lead to disorders of glucose homeostasis (hyperinsulinaemic hypoglycaemia and different forms of diabetes mellitus). Pancreatic K_ATP_, Non-K_ATP_, and some calcium channelopathies and MCT1 transporter defects can lead to various forms of hyperinsulinaemic hypoglycaemia (HH). Mutations in the genes encoding the pancreatic K_ATP_ channels can also lead to different types of diabetes (including neonatal diabetes mellitus (NDM) and Maturity Onset Diabetes of the Young, MODY), and defects in the solute carrier family 2 member 2 (*SLC2A2*) leads to diabetes mellitus as part of the Fanconi–Bickel syndrome. Variants or polymorphisms in some ion channel genes and transporters have been reported in association with type 2 diabetes mellitus.

## 1. Introduction

Ion channels and transporters are membrane-embedded proteins which play a key role in transporting ions and biomolecules across cell membranes. Although both ion channels and transporters are membrane-bound proteins, they have some differences. Ion channels are typically formed by the assembly of several different proteins and they function to allow the movement of specific ions (selectivity). The flow of ions across the channel (transmembrane flux) creates an electrical signal or action potential which is essential for the function of the particular channel [1]. Another property of ion channels is gating, which allows the channels to open and close in response to certain specific stimuli. An example of an ion channel is the adenosine triphosphate (ATP)-dependent potassium channel (K_ATP_) that allows the rapid and selective flow of K**^+^** ions across the cell membrane, and thus generates electrical signals in cells. Distinct from ion channels, some ion transporters may function as ion pumps, thereby moving ions across the plasma membrane against the electrochemical gradient.

Ion channels and transporters play essential roles in excitable cells including cardiac, skeletal and smooth muscle cells, neurons, and endocrine cells. In pancreatic beta-cells, K_ATP_ channels link the metabolic signals generated inside the cell to changes in the beta-cell membrane potential and ultimately, insulin secretion. In addition, ion channels and transporters have roles in non-excitable tissues such as the liver and skeletal muscle.

Mutations in the genes encoding the ion transporter and channel proteins lead to several diseases [2]. Disorders of glucose homeostasis (hypoglycaemia and different forms of diabetes mellitus (DM)) have been linked to several channelopathies and ion transporter defects. For example, pancreatic K_ATP_ channelopathies and defects in transporters such as GLUT1 and MCT1 are associated with different forms of hypoglycaemia and DM. In addition, Genome-Wide Association Studies (GWAS) have identified variants or polymorphisms in channel and ion transporter genes associated with type 2 diabetes mellitus (T2DM) [3,4]. For example, the non-synonymous E23K variant in the *KCNJ11* was the first robustly replicating signal to emerge as a link to T2DM.

In this review, we focus on ion channel and transporter function in relation to glucose physiology. We will firstly describe the role of ion channels and transporters in relation to insulin secretion from the pancreatic beta-cell and then describe mechanistic insights into how defects in ion channels and transporters lead to hyperinsulinaemic hypoglycemia and diabetes mellitus.

## 2. Ion Channel Defects and Hyperinsulinaemic Hypoglycaemia

Hyperinsulinaemic Hypoglycaemia (HH), is a clinically and genetically heterogeneous group of disorders characterised by inappropriate insulin secretion from the beta cell, despite low blood glucose [5,6,7]. HH is the most common cause of severe, persistent hypoglycaemia in neonates and infants with an increased risk of permanent brain damage. The most severe forms of HH are caused by mutations in the genes involved in the regulation of insulin secretion from the pancreatic beta-cell [5,6,7,8].

### 2.1. Pancreas Beta-Cell Physiology

Pancreatic beta-cells play a crucial role in the regulation of insulin secretion. ATP-sensitive potassium channels (K_ATP_) are localized in the pancreatic beta-cell membrane and allow K^+^ ion efflux across the beta-cell membrane. Glucose enters the beta cell through a facilitative glucose transporter, glucose transporter 2 (GLUT 2), and is converted to glucose-6-phosphate (G6P) by the enzyme glucokinase [9]. GLUT2 is a low-affinity transporter for glucose which allows glucose transport into the cell in proportion to the blood glucose concentrations [10]. The utilization of G6P from glycolysis generates high-energy molecules, adenosine triphosphate (ATP) and increases the ratio of ATP:ADP (adenosine diphosphate), which closes the ATP-sensitive potassium channels (K_ATP_). Closure of this channel generates a membrane potential which triggers the opening of L-type voltage-gated calcium channels allowing calcium influx into the beta cell. Increased intracellular Ca^2+^ stimulates insulin-stored granules to release insulin through exocytosis (Figure 1). Recently, it has been shown that leucine-rich repeat-containing protein 8A (LRRC8A), a subunit of volume-regulated anion channels (VRAC) enhances the beta-cell glucose sensing and insulin secretion [11].

### 2.2. ATP-Sensitive Potassium (K_ATP_) Channels and Insulin Secretion

K_ATP_ channels, members of the inwardly rectifying potassium channel family, are large octameric complexes which permit the flux of potassium ions across the cell membrane [12,13,14,15]. They are composed of four pore-forming subunits (Kir6.1 or Kir6.2, encoded by *KCNJ8* and *KCNJ11,* respectively) and four regulatory sulphonylurea receptors (SUR1, SUR2A or SUR2B) which are encoded by ATP-binding cassette subfamily C member 8 (*ABCC8*) [13,14,15]. The channel composition and its physiological roles differ based on their tissue expression, e.g., in the cardiac myocytes, pancreatic beta-cells, skeletal or vascular smooth muscles, neurons, and the kidneys [16,17,18]. One of the best-characterized potassium channels are the K_ATP_ channels in the beta-cell membrane [19]. They have an essential role in the glucose homeostasis and regulate the process of stimulated insulin release by linking the metabolism of the cell to its membrane excitability [6,19,20,21,22,23]. Defects in the genes (*KCNJ11* and *ABCC8)* encoding K_ATP_ channel proteins can lead to abnormal glucose homeostasis. Thus, the “loss-of-function” or inactivating mutations in SUR1 and Kir6.2 encoding genes lead to the most common and severe forms of HH (Figure 2) [24].

### 2.3. K_ATP_ Channel Defects (ABCC8 and KCNJ11 Gene Mutations) and HH

Defects in the subunits of the pancreatic beta-cell K_ATP_ channels due to inactivating mutations in *ABCC8* and *KCNJ11* genes are the most common genetic cause of HH [7,25]. Recessive and dominant mutations have been described in about 50% of HH patients [26,27,28]. These mutations affect the functioning of K_ATP_ channels either by altering their surface expression due to defects in trafficking or by impairing stimulation by MgADP [26]. As a result, in both cases the pancreatic beta-cell membrane will be depolarized with dysregulated insulin release, despite severely low blood glucose [29].

Recessive homozygous or compound heterozygous inactivating mutations in *ABCC8* and *KCNJ11* usually cause the most severely diffuse forms of HH, typically unresponsive to diazoxide treatment and often requiring a resection of the pancreas [28]. However, there are compound heterozygous mutations that may be milder and may respond to diazoxide [30].

Dominant inactivating mutations in *ABCC8* and *KCNJ11* genes are associated with normal K_ATP_ channel assembly and trafficking to the cell membrane, but the majority of the channel complexes have an impaired function [31]. These mutations are not so common, they are generally milder and often present later in life, although the clinical manifestations may vary in severity [32]. Dominant inactivating *ABCC8/KCNJ11* mutations typically respond to diazoxide treatment [32,33]. However, medically unresponsive forms have also been found due to a dominant negative trafficking defect [27,34].

The finding of a single, paternally-inherited K_ATP_ channel mutation with a post-zygotic loss of the corresponding maternal chromosomal region usually results in a focal adenomatous hyperplasia affecting one or more areas of the pancreas, with dysregulated insulin secretion within the lesion(s), constituting up to 30%–40% of the HH cases [35,36].

### 2.4. Non-ATP-Sensitive Potassium Channel Defects and HH (Non-K_ATP_-HH)

#### 2.4.1. Non-ATP-Sensitive (Non-K_ATP_) Potassium Channel and Insulin Secretion

The role of the non-ATP-sensitive potassium channels (non-K_ATP_) in the regulation of pancreatic insulin secretion is not well-characterized compared to K_ATP_-sensitive channels. Human beta-cells also express other non-K_ATP_ channels such as delayed rectifying (K(V)2.1/2.2) and large-conductance Ca(2+)-activated K(+) (BK) channels [24]. However, no human hypoglycaemia or diabetes phenotype has been described due to mutations in genes encoding these proteins.

#### 2.4.2. Kv11.2 Potassium Channel and HH

The voltage-gated potassium channel (Kv11.2) is encoded by the potassium voltage-gated channel subfamily H member 6 (*KCNH6*) gene. In a recent study [25] it was shown that this channel might have an important role in regulating insulin secretion in humans and mice. The study reported a multigenerational family with diabetes and neonatal HH due to mutations in the *KCNH6* gene. In addition, *KCNH6* knockout (KO) and p.P235L knockin (KI) mice were shown to mimic a phenotype of hyperinsulinaemia and hypoinsulinaemia [37] similar to that in humans. Further support for the role of *KCNH6* in potentially causing HH is provided by another study [38]. Proverbio et al. [38] performed whole exome sequencing in a group of patients with HH and identified various single nucleotide polymorphism (SNP) in the genes involved in the regulation of insulin secretion [38]. Of them, the heterozygous p.V532F mutation in the *KCNH6* has been found as a possible aetiology of HH in an Italian family [38].

#### 2.4.3. KCNQ1 Channels Mutations and HH

The *KCNQ1* gene encodes the Kv7.1 protein, a voltage-activated potassium channel α-subunit, which forms a homotetrameric channel located in the myocardium, inner ear, stomach, colon, and pancreatic beta-cells. It is critical for ion homeostasis in these tissues. The phenotype of individuals harbouring mutations in this gene includes inherited cardiac arrhythmias (long-QT syndrome (LQTS)), deafness, and gastrointestinal defects [39]. A recent report has presented evidence of HH in individuals with LQTS caused by mutations in *KCNQ1* [40]. Although the mechanism underlying the involvement of Kv7.1 in the regulation of glucose homeostasis is not entirely elucidated, the evidence suggests that it may play a role in the regulation of insulin release by regulating plasma membrane repolarization.

### 2.5. Defects in Calcium Channels and HH

#### 2.5.1. Voltage-Gated Calcium Channel and Insulin Secretion

Voltage-gated calcium (Ca^2+^) channels are ubiquitously expressed membrane channels that play a signal transducer role for membrane potential changes, leading to increased intracellular Ca^2+^ which initiate many physiological events. These include processes such as myofibrillar contraction, hormonal secretion, neurotransmission, enzyme regulation, protein phosphorylation/dephosphorylation, and gene transcription [41,42].

Human beta-cells comprise the L-type (Ca(V)1.3), P/Q-type (Ca(V)2.1), and T-type (Ca(V)3.2) calcium channels, while there are no N- or R-type Ca^2+^ channels [43]. The dominant subtype of Ca^2+^ channels involved in the regulation of glucose-induced insulin secretion are L-type (Ca(V)1.3) channels (Figure 3) [43]. Therefore, mutations in the gene that encodes for the L-type (Ca(V)1.3) channels might lead to dysregulated glucose-stimulated insulin release [43,44,45]. However, defects in the activity of T- and P/Q-type Ca^2+^ channels cause a partial (about 70%–80%) reduction in the glucose-induced insulin secretion. Membrane potential recordings suggested that L- and T-type Ca^2+^ channels participate in the action potential generation, while exocytosis of insulin-containing granules is principally triggered by Ca^2+^ influx through P/Q-type Ca^2+^ channels [43]. Although the critical role of voltage-gated calcium channels in the regulation of insulin secretion has been studied extensively, particularly in experimental studies, in clinical practice there are very few case reports with disorders in glucose metabolism as a direct consequence of defects in genes encoding the voltage-dependent calcium channels.

#### 2.5.2. CACNA1D Mutations and HH

Calcium Voltage-Gated Channel Subunit Alpha1 D (*CACNA1D)* is highly expressed in pancreatic beta-cells and encodes an L-type voltage-gated calcium channel that plays a pivotal role in the regulation of insulin secretion from the pancreatic beta-cells [6,45]. Activating germline mutations in the *CACNA1D* gene have previously been reported in patients with primary hyperaldosteronism, congenital heart defects (such as biventricular hypertrophy and ventricular septal defect), seizures, neuromuscular abnormalities, and transient diazoxide-responsive hypoglycaemia [46]. In the same report, another de novo germline mutation (p.Ile770Met) in the CACNA1D gene was found in a patient with hyperaldosteronism with the absence of HH [46]. Recently, a heterozygous de novo c.1208G>A (p.G403D) mutation has been found in the *CACNA1D* gene in a patient with diazoxide-responsive HH, heart defects (prenatal bradycardia, mild aortic insufficiency), umbilical hernia, hypermetropia, severe axial hypotonia, limb spasticity, and seizures [44]. This germline gain-of-function mutation was thought to cause an increase in the sensitivity of the L-type voltage-gated calcium channel and lead to the channel remaining open at a lower membrane potential, thereby resulting in dysregulated insulin secretion [6,46].

In electrophysiological studies, it was shown that the c.1208G>A (p.G403D) mutation leads to premature activation of the L-type voltage-gated calcium channel at a lower membrane potential, while the p.Ile770Met mutation impairs the inhibition of the channel [46]. Interestingly, *CACNA1D* mutations were also described in almost 9.3% of cases with aldosterone-producing adenoma [47]. Therefore, further investigations are required to confirm whether the *CACNA1D* gene mutations should be considered as the underlying molecular aetiology of HH.

#### 2.5.3. CACNA1C Mutations and HH (Timothy Syndrome)

Calcium Channel, Voltage-Dependent, L Type, alpha 1C Subunit (CACNA1C) is mapped on the chromosome 12p13.33 and encodes for the voltage-dependent L-type Ca-channel, Ca(V)1.2. Missense mutations of this gene cause a syndromic form of HH, Timothy syndrome, which is characterized with multi-system disorders including lethal cardiac arrhythmias, congenital heart defects, syndactyly, immune deficiency, intermittent HH, intellectual disability, autism, and autistic spectrum disorders [48].

## 3. Membrane Transporters Defects and HH

### 3.1. Monocarboxylate Transporter 1 (MCT1)

The solute carrier family 16, member 1 (SLC16A1) gene is localized on chromosome 1p13.2-1p12, spanning approximately 44 kb, and is organised as five exons intervened by four introns and encodes for a transporter protein, monocarboxylate transporter 1 (MCT1) [49]. MCT1 is a proton-linked monocarboxylate transporter that mediates the import and export of lactate, pyruvate, branched-chain oxo acids derived from leucine, valine, and isoleucine, and the ketone bodies such as acetoacetate, beta-hydroxybutyrate, and acetate through the cell membrane [50,51,52].

Lactate and pyruvate are potent insulin secretagogues. Under normal physiological conditions, lactate and pyruvate concentrations are low, and thus they are unable to trigger beta-cell insulin release. Although *SLC16A1* gene is highly expressed in other tissues, it is transcriptionally and cell-specifically silenced in the pancreatic beta-cells [53,54,55]. Dominant activating mutations in the *SLC16A1* gene promoter cause enhanced MCT1 expression in the pancreatic islets [56,57,58,59]. This increased expression leads to an increased uptake of pyruvate and its metabolism in the Krebs cycle, with a subsequently increased production of ATP and insulin secretion [56]. There are three distinct clinical conditions that have been attributed to the mutation of the *SLC16A1* gene: exercise-induced hyperinsulinaemic hypoglycaemia due to an activating mutation, erythrocyte lactate transporter defect, and MCT1 deficiency due to inactivating mutations (Figure 4) [56,57,58,59,60,61,62]. The latter two disorders do not primarily affect the glucose metabolism. Therefore, they are not in the scope of the present article.

### 3.2. Exercise-Induced Hyperinsulinaemic Hypoglycaemia due to SLC16A1 Mutations

Activating mutations in the promoter region of the *SLC16A1* gene increases MCT1 expression in beta cells, thereby increasing the influx of monocarboxylates (lactate and pyruvate) into beta cells (Figure 4) [56,57,58,59]. This, in turn, triggers inappropriate insulin release from beta cells and results in HH. Patients with a mutation in *SLC16A1* develop HH following strenuous anaerobic exercises. The affected individuals typically become hypoglycaemic within 30–45 min following a strenuous anaerobic exercise due to pyruvate and lactate accumulation, which act as insulin secretagogues [58]. Although, the majority of mutations detected in *SLC16A1* have been reported in the promoter region, recently, the first intragenic heterozygous mutation (c.556C>G, p.L186V) in *SLC16A1* was reported in a patient with HH [63].

## 4. Ion Channel and Membrane Transporter Defects in Diabetes Mellitus

### 4.1. Neonatal Diabetes Mellitus

Neonatal diabetes mellitus (NDM) is a monogenic form of diabetes which presents within the first six months of life [64,65]. NDM is autoantibody-negative type of diabetes that is caused by mutations in the genes encoding for transcription factors that regulate pancreatic development or are involved in either insulin synthesis or secretion [64,65,66]. The worldwide incidence rate has been reported at a broad range of 1/260.000 and 1/400.000 to 1/80.000-90.000 [67,68,69,70,71]. However, in the populations with a high-rate of consanguinity, the incidence of NDM has been reported at a higher rate of 1/30.000 and 1/21.000 [72,73].

NDM patients can present with a wide range of clinical manifestations as a consequence of intrauterine (low birth weight) or postnatal insulin deficiency (growth retardation, weight loss or poor weight gain, polyuria, signs and symptoms of dehydration) and in case of a delay in the diagnosis, diabetic ketoacidosis may eventually develop [64]. NDM can be transient (TNDM) or permanent (PNDM) [66,74]. TNDM remits within the first few months of life, but in the majority of these cases, diabetes relapses later in life, particularly at puberty [66,74]. However, PNDM patients require life-long therapy [66,74].

#### K_ATP_ Channel Defects (ABCC8 and KCNJ11 Mutations) and NDM

Mutations in the genes *ABCC8* and *KCNJ11* are the most frequent cause of NDM in the Western world. Interestingly, adult-onset diabetes due to potassium channel mutations can also develop or PNDM can present later in adolescence or during pregnancy [75,76,77].

Heterozygous activating *KCNJ11* mutations are the main cause of PNDM, but in rare cases can also cause TNDM (Figure 2) [78]. These mutations affect the ATP binding site to Kir6.2 (binding mutations) or indirectly reduce the ATP inhibition of the channel activity (gating mutations). Activating heterozygous or homozygous missense mutations in the *ABCC8* gene are found mainly in TNDM, but also in PNDM [76,79].

The mutations lead to transient or permanent NDM due to the over-activation of the channels insensitive to the ATP inhibition, which enhances the stimulatory effect of the MgADP and alters the beta-cell electrical activity followed by insufficient pancreatic insulin release [75,79,80,81,82]. In about 20%–30% of the patients, the potassium channel mutations (mainly Kir6.2 mutations) also affect the K_ATP_ channel complexes in multiple neurons in the brain [83] leading to deactivation of the inhibitory neurons and more severe syndromes involving neurological and psychological manifestations in addition to NDM [76,83,84,85], such as developmental delay and epilepsy (DEND syndrome), intermediate DEND syndrome without epilepsy, ataxia, muscle weakness, attention deficit hyperactivity syndrome, anxiety, autism, or sleeping disorders [84,86,87,88].

The main role of the pancreatic K_ATP_ channel activation in the pathogenesis of NDM has led to the use of sulfonylureas as a potential treatment option in 90%–95% of patients [89,90]. They stimulate insulin secretion by binding to the high-affinity SUR1 binding site, inhibiting the MgADP activation of the channels, and unmasking the ATP inhibition on Kir6.2 [79,81,90,91]. Thus, sulfonylureas can normalize the secretion of insulin and lead to improved glycaemic control in the majority of the NDM cases, especially among SUR1 patients [86,90]. However, in patients with severe channel defects and neurological deficits, sulfonylurea treatment can be partially effective or ineffective [59,61,62,90,92,93]. Based on these findings, the international guidelines (The International Society for Pediatric and Adolescent Diabetes (ISPAD) clinical guidelines) suggest immediate genetic testing after a clinical diagnosis of NDM as insulin therapy in patients with *KCNJ11* and *ABCC8* mutations can be switched to oral sulfonylureas [66,94].

### 4.2. K_ATP_ Channel Defects (KCNJ11 and ABCC8 Mutations) and MODY

Maturity-onset diabetes of the young (MODY) is а group of non-autoimmune diabetes caused by a single gene mutation that leads to a defect in the glucose-stimulated insulin secretion from the pancreatic beta-cells. It is characterized by hyperglycaemia, usually manifested before 25 years of age, and autosomal dominant inheritance among the third-generation family members. C-peptide levels are usually within the normal range, suggesting the presence of insulin secretion [94]. It is a rare form of diabetes representing less than 2% of childhood diabetes cases, with 14 different subtypes being identified up to now, with different prevalence, clinical manifestations, and treatment requirements [95].

Gain-of-function *ABCC8* and *KCNJ11* gene mutations have been reported to cause rare forms of MODY (ABCC8-MODY 12 and KCNJ11-MODY13) with variable clinical presentation, ranging from asymptomatic glucose intolerance to overt diabetes, with age [77]. Furthermore, several reports have demonstrated that some patients may develop dual phenotypes – neonatal HH and diabetes in adult-life [31,32,96,97]. The underlying mechanisms by which a mutation causes HH and diabetes later in life have been suggested as pancreatic beta-cell apoptosis due to overstimulation of insulin secretion, enhanced cell depolarization, and increased intracellular calcium influx [29,36]. Most of the ABCC8- and KCNJ11-MODY patients are responsive to sulfonylureas.

### 4.3. Other Rare Types of Monogenic Diabetes due to Membrane Transporter Defects

#### 4.3.1. Glucose Transporter 2 (GLUT 2) Deficiency and Diabetes Mellitus (Fanconi-Bickel syndrome; Glycogen Storage Disease Type XI)

Glucose transporters (GLUTs) are members of a large group of the solute carrier transporter superfamily, encoded by the *SLC2A* genes [98,99]. There are 4 main subtypes of GLUTs expressed in different tissues and involved in the organ-specific glucose transport. GLUT-1 is involved in the basal non-insulin-induced glucose uptake into many cells. GLUT-2 plays a role in the glucose sensing mechanism by mediating facilitative glucose transport into the beta cells (Figure 1). GLUT-3 is mainly involved in the non-insulin-mediated glucose uptake into brain neurons and placenta. And finally, GLUT-4 is mostly expressed in the muscle and adipose tissue and thereby mediates the peripheral action of insulin.

Solute carrier family 2 member 2 (SLC2A2), also known as the *GLUT2* gene, consists of 11 exons and 10 introns spanning approximately 30 kb, and encodes for a glucose transporter known as GLUT2 [100]. GLUT2 is a facilitative glucose transporter, expressed in the liver, pancreatic beta-cell, renal tubular, and intestinal epithelial cells. GLUT2 mediates the passive transport of intracellular glucose and galactose across the basolateral membrane and down the concentration gradient [101]. After a carbohydrate-rich feeding, GLUT2 transports glucose and galactose into the hepatocytes and plays a role in the release of glucose from the liver during the fasting state [102]. GLUT2 is located on the apical membrane and mediates the absorption of simple sugars as a result of its temporary expression on the apical membrane of the intestinal mucosa, in a process that is independent of sodium-glucose transporter 1 (SGLT1 or SLC5A1). GLUT2 is involved in glucose transport in the beta cells and thereby regulates glucose-stimulated insulin secretion. Autosomal-recessive mutations of the *SLC2A2* gene lead to Fanconi–Bickel syndrome (FBS), while autosomal-dominant mutations result in non-insulin dependent DM [103,104,105,106,107,108].

Fanconi–Bickel syndrome is a rare autosomal recessive disorder of carbohydrate metabolism leading to the accumulation of glycogen in the liver and kidneys. Homozygous or compound heterozygous mutations of the *SLC2A2* gene are responsible for the molecular basis of FBS [102,106,107]. Patients may present with hepatomegaly, glucose and galactose intolerance, proximal tubular nephropathy, severe growth retardation, and rickets due to the accumulation of glycogen [106,107]. The mechanism of glycogen accumulation is a defective transport of glucose from the intracellular compartment to the extracellular compartment during glycogenolysis, due to GLUT2 deficiency. This causes a markedly elevated intracellular glucose, and thereby, inhibition of glycogenolysis. These patients suffer from postprandial hyperglycaemia, fasting hypoglycaemia, and DM. Hyperglycaemia and hypergalactosemia occur after feeding in patients with FBS as a result of the defective transport and decreased uptake of monosaccharides by the liver, and hyperglycaemia is further aggravated by decreased glucose-stimulated insulin secretion in pancreatic beta-cells. Fasting hypoglycaemia develops due to an impaired glucose export in the hepatocytes when peripheral glucose sources are depleted [102]. Glycated haemoglobin (HbA1c) is usually within normal range due to recurrent hypoglycaemia episodes. Fasting hypoglycaemia and postprandial hyperglycaemia have been shown to improve over time [109].

Proximal renal tubular dysfunction is characterized by glucosuria, phosphaturia, generalized aminoaciduria, and urinary bicarbonate loss, or may cause refractory rickets, which can be the presenting feature of FBS in infancy [107,110]. Renal glucosuria, which may occur at the blood glucose level of below renal glucosuria threshold (180 mg/dl), results from the impaired glucose reabsorption at the proximal renal tubular basolateral membrane.

#### 4.3.2. Thiamine-Responsive Megaloblastic Anaemia (TRMA) and Diabetes Mellitus

The Solute Carrier Family 19 Member 2 *(SLC19A2)* gene, located on chromosome 1q23.3, is composed of six exons and encodes for a high-affinity thiamine transporter 1 protein (THTR-1) containing 497 amino acids and 12 transmembrane domains [111,112,113,114,115]. *SLC19A2* is expressed in various tissues, such as bone marrow, liver, colon, small intestine, pancreas, brain, retina, heart, skeletal muscle, kidney, lung, placenta, lymphocytes, and fibroblasts [111]. Thiamine plays a role in carbohydrate metabolism and energy production. There are two main thiamine transporter proteins which have high-affinity to thiamine located in the intestine, thiamine transporter 1 (THTR-1) which is encoded by the *SLC19A2* gene, and thiamine transporter 2 (THTR-2) which is encoded by the *SLC19A3* gene (Figure 5) [111,116,117]. Most tissues express both genes, although cochlear inner hair cells, pancreatic islet cells, and erythropoietic precursor cells express only *SLC19A2*. At high concentrations, thiamine is transported across the cell membrane via passive diffusion [118,119,120]. However, the active transport of thiamine through transporter mechanisms is impaired in thiamine-responsive megaloblastic anaemia (TRMA) syndrome resulting in intracellular thiamine deficiency. There are THTR-2-mediated compensatory mechanisms activated in most cells. However, three cell lines: cochlear inner hair cells, pancreatic islet cells, and erythropoietic precursor cells are THTR-1-dependent, and therefore they are mostly affected by thiamine deficiency.

Thiamine-responsive megaloblastic anaemia (TRMA), also known as Roger’s Syndrome, is an autosomal recessive disorder characterized by early-onset non-autoimmune DM, megaloblastic anaemia, and sensorineural deafness (SND) [113,121,122]. Other well-defined clinical features are congenital heart disease, arrhythmias, visual disturbances, retinal degeneration, optic atrophy, aminoaciduria, short stature, situs inversus, polycystic ovarian syndrome, stroke, and neurological disorders [123]. The disease can manifest at any time from infancy to adolescence, while cardinal manifestations may not be apparent at the initial presentation, but develop later over time.

As the TRMA is inherited in an autosomal recessive manner, it is more frequent in the consanguineous pedigrees [112]. However, TRMA due to compound heterozygous mutations has also been reported in non-consanguineous families [124,125]. To date, 51 mutations have been described in the *SLC19A2* gene, the majority of which are missense and nonsense mutations (http://www.hgmd.cf.ac.uk/ac/gene.php?gene=SLC19A2). Most mutations in the *SLC19A2* gene result in an abnormally truncated, nonfunctional THTR-1 protein. It has been demonstrated that some mutations cause single amino acid changes in THTR-1, resulting in an abnormal folding of the protein, or making them unable to traffic the transport protein to the cell surface which results in a defect in the intracellular transport of thiamine by THTR-1.

A high level of thiamine is required for the normal exocrine and endocrine functions of the pancreas [126]. Impaired insulin secretion has been demonstrated in the islet cells of thiamine-deficient mice [127]. Stagg et al. suggested that the primary disorder in TRMA is the deficiency of high-affinity thiamine transporters as low intracellular thiamine results in cellular death through apoptosis in the patient’s fibroblasts [128]. The insulin requirement physiologically increases in puberty, and pancreatic beta-cell reserves decline, possibly due to apoptosis, while the residual reserve fails to meet the insulin requirement in puberty [129]. In TRMA, thiamine replacement does not store insulin secretion at puberty. Therefore, these patients may have an insulin requirement despite thiamine replacement.

The goal of treatment for TRMA is symptom relief, and thiamine (vitamin B1) supplementation at pharmacological doses (50-100 mg/day) corrects haematological and endocrine manifestations, although neurological symptoms do not respond to the therapy [121,130]. Although the therapy corrects anaemia and hyperglycaemia, there are reports of patients requiring insulin therapy and regular blood transfusions in adulthood [115,131].

## 5. Variants in Ion Channel and Transporters Genes Associated with Type 2 Diabetes Mellitus

T2DM is rapidly increasing throughout the world and is characterized by hyperglycemia caused by defects in insulin secretion, insulin action, or both. The underlying genetic mechanisms of T2DM involve genetic and environmental factors. Genome-Wide Association Studies (GWAS) have identified a significant number of different loci, mostly in the non-coding regions of genes implicated in type 2 diabetes [3,4]. The non-synonymous E23K variant in the *KCNJ11* was the first robustly replicating signal to emerge as a link to T2DM [132].

Mechanistic studies suggest that the E23K variant may have a diabetogenic effect by increasing the K_ATP_ channel activity in response to changes in the level of long-chain acyl CoAs which increase during fasting [133].

Two GWAS studies have identified polymorphisms in intron 15 and 11 of the potassium voltage-gated channel, KQT-like subfamily member 1 (KCNQ1) in association with T2DM [134,135]. The increased susceptibility for developing T2DM linked to polymorphisms in the *KCNQ*1 gene is likely to be caused by a reduction in insulin secretion. The pore-forming alpha subunit of the voltage-gated K+ channel (KvLQT1) (encoded by *KCNQ1*) and the regulatory beta subunit ISK (encoded by potassium channel, voltage-gated, ISK-related subfamily, member 1: *KCNE1* gene) co-assemble to form the I_(KS)_ potassium channel in the pancreas. Thus, there is a possibility that *KCNQ1* polymorphisms alter the role of the I_(KS)_ potassium channel, leading to the decreased insulin secretion [136].

The *SLC30A8* gene encodes a zinc transporter family member 8 (ZnT8) in pancreatic beta-cells and is responsible for the accumulation of zinc in secretory granules. A non-synonymous polymorphism in *SLC30A8* is associated with the risk of developing T2DM [137]. The precise mechanisms by which polymorphisms in the *SLC30A8* are associated with T2DM are not fully understood but might involve a reduction in insulin secretion, increased insulin clearance, or changes in reactive oxygen species.

## 6. Conclusions

Ion channel and transporter defects lead to HH and various forms of DM. Defects in the pancreatic K_ATP_ channels can lead to HH, NDM (both transient and permanent), MODY, and variants in KCNJ*11* and *KCNQ1* channel genes are associated with T2DM. Understanding the molecular mechanisms of HH and different types of diabetes due to the ion channel and transporter defects has provided unique insights into the role of these proteins in normal physiology, especially in the pancreatic beta-cell. Further studies are required to develop pharmacological agents which can target defects in channel proteins and treat conditions such as severe HH. In the case of NDM, understanding the role of the pancreatic K_ATP_ channels in insulin secretion has transformed the lives of patients as they can now be treated with oral sulphonylureas.

## Figures and Tables

**Figure 1 ijms-20-02590-f001:**
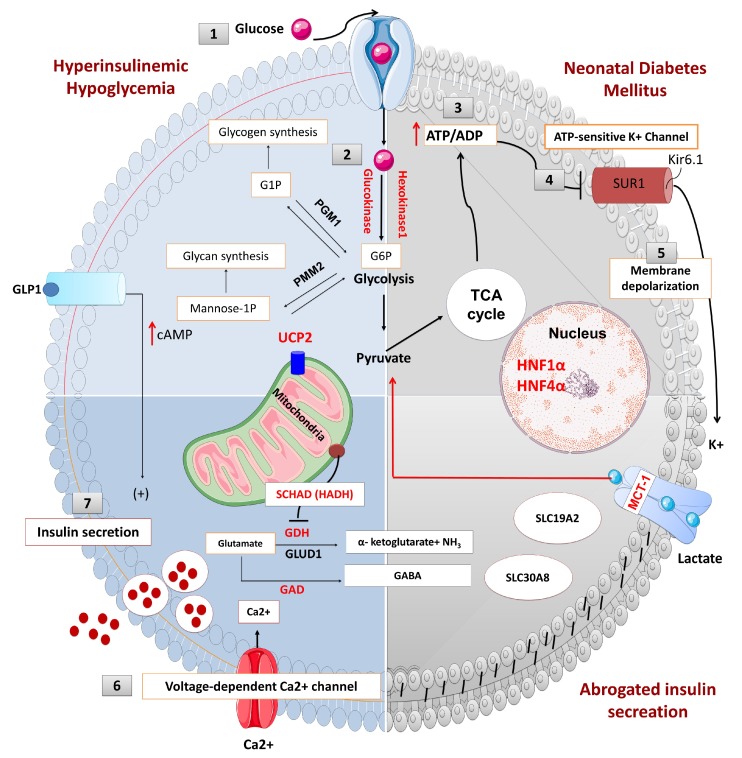
The physiology of insulin secretion from pancreatic beta-cells. Glucose enters into the beta cells through facilitative transport mediated by glucose transporter 2 (GLUT 2) (1) and is converted to glucose-6-phosphate by the enzyme glucokinase (2). Generation of high energy molecules such as adenosine triphosphate (ATP), leads to an increase in the ratio of ATP/ADP (adenosine diphosphate) (3). Elevated ATP/ADP ratio closes the ATP-sensitive potassium channel (K_ATP_). The Kir6.2 subunits of the K_ATP_ channels are responsible for K^+^ ion efflux from the pancreatic beta-cell, and thus maintains a steady state membrane potential (4). The closure of the K_ATP_ channels results in a depolarization of the pancreatic beta-cell membrane and the activation of voltage-gated calcium channels located on the beta-cell membrane (5). Calcium enters into beta cells through these voltage-gated calcium channels (6), and the increase in intracellular calcium triggers secretory granule exocytosis and insulin release (7). In congenital hyperinsulinism, defects in the K_ATP_ channel or energy metabolism leading to prolonged closure of K_ATP_ channels, create a membrane potential between the inner and outer sites of the beta cell (depolarization) which is followed by the opening of the voltage-gated calcium channel and calcium influx. Constantly closed K_ATP_ channels or uncontrolled ATP generation uncouples the insulin secretion from the blood glucose level and causes inappropriate insulin secretion, despite low blood glucose. On the contrary, in neonatal diabetes mellitus (NDM), the K_ATP_ channel remains open and a constant efflux of K^+^ ion hyperpolarizes the beta cell. Lack of depolarization does not allow for voltage-gated calcium channel opening and insulin secretion despite elevated blood glucose. (Ca^2+^: calcium ions, cAMP: cyclic adenosine monophosphate, G1P: glucose-1-phosphate, G6P: glucose 6-phosphate, GABA: γ-aminobutyric acid, GAD: glutamate decarboxylase enzyme, *GLUD1*: glutamate dehydrogenase 1, GDH: glutamate dehydrogenase, GLP1: glucagon-like peptide 1, GLUT2: glucose transporter 2, HADH: hydroxy acyl-CoA dehydrogenase, HNF1α: hepatocyte nuclear factor 1α, HNF4α: hepatocyte nuclear factor 4α, K^+^: potassium, Kir6.2: inward rectifier potassium channel 6.2, MCT1: monocarboxylate transporter 1, NH3: ammonia, PGM1: phosphoglucomutase 1, PMM2: phosphomannomutase 2, SUR1: sulfonylurea receptor 1, TCA: tricarboxylic acid, UCP2: mitochondrial uncoupling protein 2).

**Figure 2 ijms-20-02590-f002:**
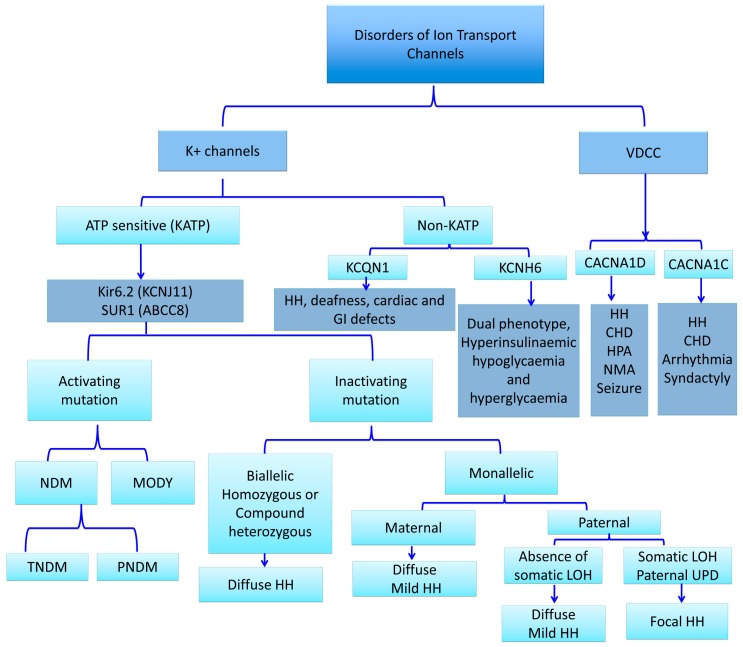
Summary of ion transport channel defects and related disorders (K+: Potassium, VDCC: Voltage dependent calcium channels, K_ATP:_ ATP-sensitive potassium channels, KIR6.2: Inward rectifier potassium channel subunit 2, KCNJ11: Potassium Voltage-Gated Channel Subfamily J Member, SUR1: Sulphonylurea receptor 1, ABCC8: ATP-binding cassette subfamily C member 8, MODY: Maturity-onset diabetes of the young, CACNA1D: Calcium Voltage-Gated Channel Subunit Alpha1 D, CACNA1C: Calcium Channel, Voltage-Dependent, L Type, alpha 1C Subunit, KCNH6: Potassium voltage-gated channel subfamily H member 6, NDM: Neonatal diabetes mellitus, TNDM: Transient NDM, PNDM: permanent NDM, HH: Hyperinsulinaemic Hypoglycemia, CHD: Congenital heart diseases, HPA: Hyperaldosteronism, NMA: Neuromuscular abnormalities, LOH: Loss of heterozygosity, UPD: Uniparental isodisomy).

**Figure 3 ijms-20-02590-f003:**
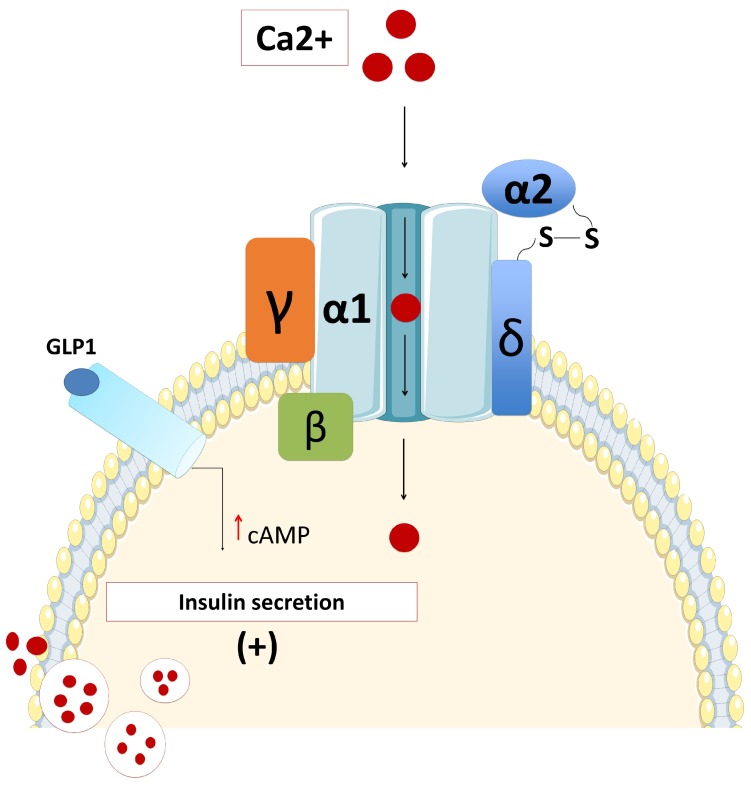
Opening of the “Voltage-dependent calcium channel VDDC” located on the beta-cell membrane increases calcium influx and increased intracellular calcium trigger beta-cell insulin secretion through exocytosis. GLP1 contribute to the exocytosis process by increasing the cAMP (GLP1: Glucagon like peptide 1, cAMP: Cyclic adenosine monophosphate).

**Figure 4 ijms-20-02590-f004:**
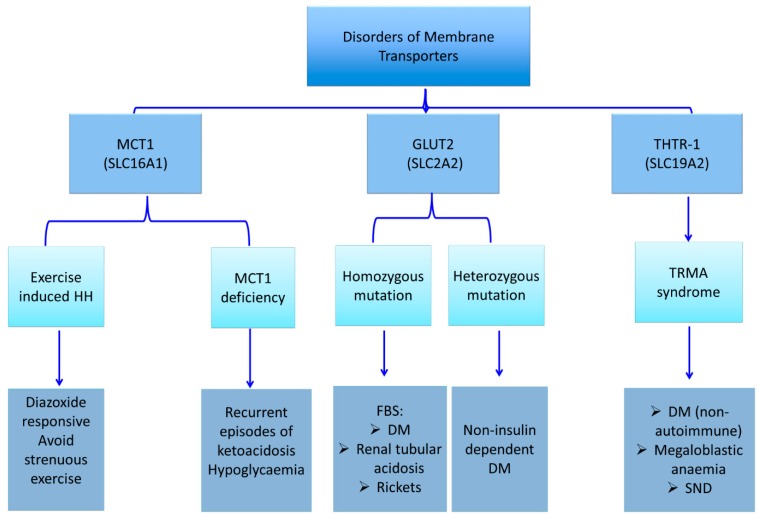
Summary of membrane transporters defects and related disorders (MCT1: monocarboxylate transporter 1, SLC16A1: Solute carrier family 16, member 1, GLUT 2: Glucose transporter 2, SLC2A2: Solute carrier family 2 member 2, THTR-1: Thiamine transporter 1 protein, SLC19A2: The Solute Carrier Family 19 Member 2, HH: Hyperinsulinaemic hypoglycaemia, TRMA: Thiamine-responsive megaloblastic anaemia, DM: Diabetes Mellitus, FBS: Fanconi Bickel syndrome, SND: Sensorineural deafness).

**Figure 5 ijms-20-02590-f005:**
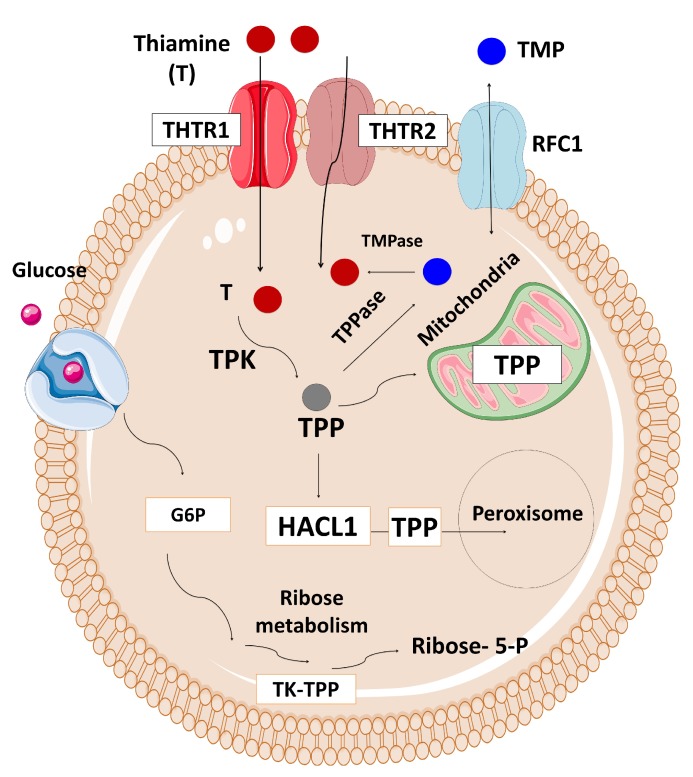
The role of thiamine transporters, THTR-1 in glucose metabolism. (THTR-1: Thiamin transporter 1, THTR-2: Thiamin transporter 2, RFC1: Reduced folate carrier 1, TMP: Thiamine monophosphate, TMPase: Thiamine monophosphatase, HACL1: 2-Hydroxyacyl-CoA Lyase 1, TPP: Thiamine pyrophosphate, TPPase: Thiamine pyrophosphatase, TK-TPP: Transketolase-Thiamine pyrophosphate, TPK: Thiamine pyrophosphokinase, TDP: Thiamine diphosphate).

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
