# Peer review of "Ion Transporters, Channelopathies, and Glucose Disorders"

_ijms, 2019, doi:10.3390/ijms20102590_

Round 1
Reviewer 1 Report
This is a very difficult-to-follow review about the involvement of ion channels and transporters in insulin secretion by the beta cell.
I have many problems to follow the review. The titles of the different sections are not organized. The figures are very complex and are not linked to the position of the text. In the introduction, the should clarify what is the review about.
Furthermore, I have found many mistakes. As an example, just in the introduction: Glut2 is not a high affinity glucose transporter, but rather a low affinity glucose transporter (Km is in the mM range). Ion transporters not only move ions against the electrochemical gradient.
Finally, the recent discoveries of the role of VRAC chloride channels in insulin secretion is not mentioned at all in the text.
Author Response
Reviewer 1
Reviewer comment: This is a very difficult-to-follow review about the involvement of ion channels and transporters in insulin secretion by the beta cell. I have many problems to follow the review. The titles of the different sections are not organized. The figures are very complex and are not linked to the position of the text. In the introduction, the should clarify what is the review about.
Authors’ reply: We thank the reviewer for kindly attention. We have re-organized the titles and subtitles of the relevant sections. We also revised the figures and expanded the figure legends to better understand.
Reviewer’s comment: Furthermore, I have found many mistakes. As an example, just in the introduction: Glut2 is not a high affinity glucose transporter, but rather a low affinity glucose transporter (Km is in the mM range).
Authors’ reply: We thank the reviewer for kindly attention. We corrected such mistake on the text as the reviewer suggested. We also added a new reference for this statement (ref 10)
Reviewer’s comment: Ion transporters not only move ions against the electrochemical gradient.
Authors’ reply: We thank the reviewer for kindly attention. We changed the above statement as the reviewer suggested (Line 38-39).
Reviewer’s comment: Finally, the recent discoveries of the role of VRAC chloride channels in insulin secretion is not mentioned at all in the text.
Authors’ reply: We thank the reviewer for the kindly suggestion. We added a short statement about recently discovered VRAC chloride channel and their role on insulin secretion. But since our scope was the channel physiology, disorders and clinical consequences we could not find any clinical report indicating glucose disorder in humans due to a defect in the VRAC chloride channels.
Reviewer 2 Report
Overall, the manuscript is of high quality. A glaring error appears on line 38-39. Ion transporters cannot violate the laws of thermodynamics. They do not move ions against the electrochemical gradient. Rather, ion pumps can create ion gradients that are used by ion channels and some types of ion transporters. This sentence must be changed.
Author Response
Reviewer 2
Reviewer’s comment: Overall, the manuscript is of high quality. A glaring error appears on line 38-39. Ion transporters cannot violate the laws of thermodynamics. They do not move ions against the electrochemical gradient. Rather, ion pumps can create ion gradients that are used by ion channels and some types of ion transporters. This sentence must be changed.
Authors’ reply: We thank the reviewer for kind attention. We changed the above statement as the reviewer suggested.
Round 2
Reviewer 1 Report
The authors incorporated into the manuscript some of my minor comments. However, I still think that it is difficult to follow since there are many titles and subtitles. I think that the best way is that you put numbers to every section. It is: 1. Introduction 2. Ion channel defects and hyperinsulinaemic hypoglycaemia. 2.1 Pancreatic katp channels and beta cell physiology.....2.4 Non-ATP sensitive K channels defects and HH. 2.4.a Non-ATP sensitive potassium channel...
and so on. Otherwise, I am really confused to know each part to what part belongs.
Author Response
We have now numbered all the relevant sections as recommedned by this reviewer.